# The Abbreviated Maslach Burnout Inventory Can Overestimate Burnout: A Study of Anesthesiology Residents

**DOI:** 10.3390/jcm9010061

**Published:** 2019-12-26

**Authors:** Wan Yen Lim, John Ong, Sharon Ong, Ying Hao, Hairil Rizal Abdullah, Darren LK Koh, Un Sam May Mok

**Affiliations:** 1Department of Anesthesiology, Singapore General Hospital, Singapore 169608, Singapore; sharon.ong.g.k@singhealth.com.sg (S.O.); hairil.rizal.abdullah@singhealth.com.sg (H.R.A.); mok.un.sam@singhealth.com.sg (U.S.M.M.); 2Department of Engineering, University of Cambridge, Cambridge CB2 1PZ, UK; 3School of Medicine, National University of Singapore, Singapore 119077, Singapore; 4Health Services Research Unit (HSRU), Singapore General Hospital, Singapore 169608, Singapore; hao.ying@sgh.com.sg

**Keywords:** burnout, anesthesiology, trainees in anesthesia, anesthetists, Maslach Burnout Inventory, abbreviated Maslach Burnout Inventory, residents

## Abstract

The Maslach Burnout Inventory for healthcare professionals (MBI-HSS) and its abbreviated version (aMBI), are the most common tools to detect burnout in clinicians. A wide range in burnout prevalence is reported in anesthesiology, so this study aimed to ascertain which of these two tools most accurately detected burnout in our anesthesiology residents. The MBI-HSS and aMBI were distributed amongst 86 residents across three hospitals, with a total of 58 residents completing the survey (67.4% response rate; 17 male and 41 female). Maslach-recommended cut-offs for the MBI-HSS and the aMBI with standard cut-offs were used to estimate burnout prevalence, and actual prevalence was established clinically by a thorough review of multiple data sources. Burnout proportions reported by the MBI-HSS and aMBI were found to be significantly different; 22.4% vs. 62.1% respectively (*p* < 0.0001). Compared to the actual prevalence of burnout in our cohort, the MBI-HSS detected burnout most accurately; area under receiver operating characteristic of 0.99 (95% confidence interval (CI): 0.92–1.0). Although there was a good correlation between the MBI-HSS and aMBI subscale scores, the positive predictive value of the aMBI was poor; 33.3% (95% CI:27.5–39.8%), therefore caution and clinical correlation are advised when using the aMBI tool because of the high rates of false-positives.

## 1. Introduction

Burnout is a work-related syndrome characterized by depersonalization, emotional exhaustion, and low personal accomplishment, leading to detrimental professional and personal consequences. Since May 2019, burnout has been recognized as an “occupational phenomenon” in the 11th revision of the International Classification of Disease published by the World Health Organization. Globally, it is estimated that 30–50% of clinicians experience symptoms of burnout [1], but it is unclear whether these symptoms translate into burnout syndrome. Nonetheless, undetected and unaddressed, clinician burnout can result in poorer patient satisfaction, impaired professionalism and communication, depression and suicidal ideations, professional errors and near misses, which may impact patient outcomes [1]. Furthermore, clinicians experiencing burnout may develop depression, sleep disturbances, alcoholism, musculoskeletal disorders, hypertension and ischemic heart disease [2,3,4,5].

Anesthesiology is a stressful specialty and a self-perceived lack of personal accomplishment may be more common because efforts for safe anesthesia are often not acknowledged [6,7,8]. Indeed, recent studies have suggested that burnout is common in anesthesiology residents and trainees [9,10,11]. In a national US study, the prevalence of burnout, distress and depression in 5295 anesthesiology residents were reported as 51%, 32% and 12%, respectively [9]. Addressing burnout in anesthesiology residents early, reduces the detrimental effects on long-term physical and mental health, thereby prolonging the longevity and well being of a skilled workforce [12,13]. There are three anesthesiology residency training programs in Singapore that come under the purview of the Accreditation Council of Graduate Medical Education International (ACGME-I). Of these, the SingHealth Anesthesiology Residency Program (SHARP) is the largest training program spanning three hospitals, therefore, establishing the prevalence of burnout in SHARP would be a good indication of the prevalence of burnout among anesthesiology residents in Singapore.

The 22-question Maslach Burnout Inventory for healthcare professionals (MBI-HSS) contains three subscales, which assess emotional exhaustion (EE), depersonalization (DP), and a sense of low personal achievement (LPA). It is the most commonly used and validated tool to detect burnout in clinicians so it was considered as the gold standard [14,15]. The abbreviated version of the MBI-HSS (aMBI) contains 12 of the 22 questions from the MBI-HSS but utilizes different cut-offs to define burnout [16]. It was adopted as a second tool to measure burnout because it enabled the direct comparison of results between this study to the many other studies of anesthesiology residents which used the aMBI. Gabbe et al. (2002) previously validated the aMBI, reporting that it closely mirrored the MBI-HSS [16]. As the aMBI is easier to administer and improves survey response rates, it has been widely adopted in burnout studies within anesthesiology [9,11,17,18,19,20,21]. Moreover, discrepancies between the MBI-HSS and aMBI have not been previously reported in the literature.

The primary objective of this study was to determine the prevalence of burnout in anesthesiology residents within the SHARP residency program. The secondary objective was to determine which of the two tools, the MBI-HSS or the aMBI, detected burnout most accurately in our cohort, to facilitate future studies of burnout. Since both tools have been extensively utilized in literature, we hypothesized the readouts from the MBI-HSS and the aMBI would be relatively similar.

Surprisingly, this study found a very significant difference in burnout prevalence reported by the MBI-HSS and the aMBI. An in-depth review of data from multiple sources by a research committee concluded the actual prevalence of burnout was low, similar to that detected by the MBI-HSS. Burnout prevalence detected by the aMBI was found to be three-fold higher than the actual prevalence. Herein, we report our analyses of the MBI-HSS and aMBI as diagnostic tools for burnout, the prevalence of burnout, and perceived stressors by anesthesiology residents in Singapore.

## 2. Experimental Section

### 2.1. Study Design

Observational, quantitative and qualitative, multi-center study.

#### 2.1.1. Study Participants

Anesthesiology residents from the largest Anesthesiology Residency Program in Singapore based in three affiliated teaching hospitals within Singapore: the Singapore General Hospital, Changi General Hospital, and KK Women’s and Children’s Hospital.

#### 2.1.2. Data Collection

A three-part hardcopy survey was distributed to 86 anesthesiology residents within the SingHealth Anesthesiology Residency Program between June 2017 and November 2017, with a total of 58 residents completing the surveys (67.4% response rate).

The first part of the survey collected information regarding the residents’ demographics, the second part comprised of the MBI-HSS, and the third part collected information on self-perceived stressors and coping strategies adopted by anesthesiology residents. The demographic details collected were age, gender, marital status, number of children, local or overseas medical school, and year of residency. The survey was administered in English, and one reminder email was sent one month before the end of data collection.

The second part of the survey contained the MBI tools: the MBI-HSS and the aMBI. The MBI-HSS comprised 22 questions across 3 subscales; Emotional Exhaustion (EE)—9 questions, Depersonalization (DP)—5 questions, and self-perceived lack of Personal Accomplishment (PA)—8 questions. Responses were scored on a six-point Likert scale (ranging from 0 = never to 6 = everyday) for each subscale, and tabulated into three tiers (low, moderate or high) based on the reference ranges provided with the MBI-HSS: for EE, low (0–16), moderate (17–26) and high (≥27); for DP, low (0–6), moderate (7–12) and high (≥13), and finally, for PA, low (≤31), moderate (32–38) and high (≥39). Burnout was defined by the updated Maslach-recommended criteria of “high EE and high DP” or “high EE and low PA” [22]. To allow for the direct comparison of burnout rates between this study and other studies of anesthesiology residents, the aMBI score was used as an alternative tool to measure burnout prevalence. The aMBI scores were calculated using a weighted scoring of the twelve questions within the MBI-HSS that constitute the aMBI. Previously, creators of the aMBI established reference ranges for the subscale as EE: low (0–18), moderate (19–26) and high (≥27); for DP: low (0–5), moderate (6–9) and high (≥10), and for PA: low (0–33), moderate (34–39) and high (≥40) [16]. For the aMBI, burnout is defined as moderate scores in two or more subscales [16]. Furthermore, Gabbe et al. (2002) had previously validated the aMBI and reported that it could substitute the MBI-HSS [16].

In the third part of the survey, residents were asked to rank stressors they perceived to be most significant in contributing to burnout, which encompassed four domains: time management, training-related stressors, work-related stressors, and non-work–related stressors. Each domain comprised several questions, and respondents selected binary answers, “yes” or “no”. Four questions allowed the entry of free text. Examples of burnout stressors included long working hours, specialty examinations and performance appraisals, residency requirements (e.g., logbook/research), personal stressors such as family commitments, and financial burden. Respondents were also asked about their coping strategies, which included physical activities (e.g., sports, yoga, meditation), engaging social support (e.g., family or friends), and behavioral habits (e.g., alcohol intake, smoking).

Before distribution, a test group demonstrated the overall internal consistency of the survey was good with a Cronbach alpha value of 0.89 (95% confidence interval (CI): 0.85–0.93). All residents in anesthesia were included, and participation was voluntary. No financial incentives were offered for responses. All responses were kept anonymous, and incomplete surveys were excluded. None of the authors herein participated in the survey.

### 2.2. In-Depth Review of Multiple Data Sources

A research committee reviewed anonymized data gathered from (i) resident-resident representative meetings, (ii) Program Evaluation Committee (PEC) meetings, (iii) Clinical Competency Committee (CCC) meetings, (iv) resident surveys on their residency (ACGME-I), (v) sickness and absenteeism records and (vi) multi-source feedback on clinical care, professionalism and communication from July 2017 to June 2018, which corresponded to the residency academic year. Data were scrutinized for signs of burnout: patient feedback, resident feedback, peer feedback, supervisor feedback, clinical errors, staff incidents, sickness and absenteeism, violation of duty hours, recurrent failure in postgraduate examinations, failure to progress in training, and resignations. Residents were also asked, “Do you feel burned out from your work?” and all answers, including comments, were recorded anonymously. These data were assessed qualitatively to evaluate the prevalence of burnout. Any resident identified or self-reported as being in difficulty or burnt out during this process was provided the option of additional support through already established mechanisms within the training program to help residents in difficulty. 

### 2.3. Ethics

The study was approved by the Singhealth Institutional Research Board ethics committee (CIRB reference 2016/3026) before the commencement of the study.

### 2.4. Statistical Analysis

The Shapiro–Wilk test was used to test for normality of data [23]. Parametric data were reported as median ± standard deviation (SD), and non-parametric data were reported as median with inter-quartile range (IQR). Parametric data were compared using Student’s *t*-test, and non-parametric data were compared using the Mann-Whitney U test where appropriate. The chi-square test was used to compare the proportion of burnout and sub-group distributions in the demographic data. The Fisher exact test was used when *n* ≤ 5 for any cell. Posthoc analysis using Bonferroni correction was applied if a significant *p*-value was obtained from multiple hypothesis testing in sub-group analyses. Two-tailed tests were used in all tests, and the alpha was set at 0.05 (95% confidence interval).

Pearson’s correlation was used to study the correlation between the MBI-HSS and the aMBI subscales, and the correlation coefficient was reported. The accuracy of the aMBI tool was evaluated using 2 × 2 tables and the area under receiver operating characteristic (AUROC) scores. Logistic regression was used to determine if a relationship between demographic variables and burnout existed. Ordinal data were coded as JR1 = 0, JR2 = 1, JR3 = 2, JR4 = 3, SR1 = 4, SR2 = 5; and age < 25 years = 0, age 25–28 years = 1, age 29–32 years = 2, age 33–36 years = 3, and age > 36 years = 4. Categorical data was coded as male = 1, female = 0; Yong Loo Lin (Y.L.L.) undergraduate medical school = 0, Duke-NUS postgraduate medical school = 1, foreign medical schools = 2; burnout = 1, non-burnout =0. Logistic regression was also used to determine if a relationship between stressors and burnout existed. All variables were coded as categorical variables: Yes = 1, No = 0; and burn out = 1, non-burnout = 0.

Statistical analyses were performed using MedCalc for Windows, version 19.1 (MedCalc Software, Ostend, Belgium), and R 3.4.2 (R Core Team 2017. R: A language and environment for statistical computing. R Foundation for Statistical Computing, Vienna, Austria).

## 3. Results

### 3.1. Demographics of Respondents and Physicians in the Anesthesiology Residency Program

Physician response rates in unpaid, voluntary surveys are commonly very low (20–50%), posing a well-recognized problem in research [24,25,26,27,28]. This survey had a response rate of 67.4% (58/86), and a good representation of our cohort was achieved (Table 1). Furthermore, the chi-square test did not show any statistical differences between the entire cohort of residents and respondents in the survey at different stages of training, gender, age groups, and country of graduation. Coincidentally, there were significantly more female than male anesthesiology residents in our residency program (67.4% vs. 32.6%, *p* < 0.01), and in the respondents who completed the survey (70.7% vs. 29.3%, *p* < 0.01).

Using the MBI-HSS, the prevalence of burnout was found to be 22.4% (13/58) in comparison to a prevalence of 62.1% using the aMBI; this difference was statistically significant (*p* < 0.0001). To understand the differences in results and ascertain which of these results were accurate, the aMBI tool was analyzed further.

### 3.2. The aMBI Subscale Scores Derived from Weighted Scoring Correlated Accurately with the MBI-HSS

The aMBI contains the same three subscales as the MBI-HSS, namely EE, DP, and PA but the fundamental difference is that the scores are calculated from select questions and multiplied by a factor (weighted scoring) to obtain a final subscale score. More specifically in the aMBI, the EE score is calculated by the sum of questions 1, 2, 8, 13, and 20, then multiplied by a factor of $\frac{9}{5}$, the DP score is calculated by the sum of questions 5, 10, and 11, then multiplied by a factor of $\frac{5}{3}$, and PA score is calculated by the sum of questions 7, 9, 18, and 19, then multiplied by a factor of 2. In contrast, the subscale scores in the MBI-HSS are derived by the addition of all question scores within the subscale.

Pearson’s correlation (Figure 1) demonstrated that despite the omission of ten questions from the MBI-HSS, there was still excellent correlation between aMBI and the MBI-HSS subscale scores, EE: *r* = 0.96 (95% CI: 0.93–0.98), DP: *r* = 0.97 (95% CI: 0.96–0.99) and PA: *r* = 0.92 (95% CI: 0.86–0.95).

### 3.3. Differences between the Abbreviated Maslach Burnout Inventory (aMBI) and Maslach Burnout Inventory for Healthcare Professionals (MBI-HSS) Stemmed from Variations in Subscale Cut-Off Values and Definitions of Burnout

Despite the excellent correlation between subscale scores, there were differences in burnout detection as a result of different subscale cut-off values in the aMBI tool and the criteria used to define burnout. In the aMBI, even though the cut-off for moderate EE scores was set higher than the MBI-HSS, less-stringent DP and PA cut-offs markedly increased the number of anesthesiology residents with moderate scores in the latter two domains. Furthermore, the aMBI defines burnout as the presence of two or more moderate scores [16], with the combination of these factors significantly increasing the number of burnout cases reported. Table 2 compares and contrasts the differences in subscale scores that determine the risk of burnout, low, medium, or high, and the proportion of residents reported with burnout symptoms.

### 3.4. An In-Depth Review of Multiple Data Sources Confirmed the Prevalence of Burnout Was Low

Burnout is classically characterized by high levels of emotional exhaustion, depersonalization and a low sense of personal achievement. Diagnostic tools often assess the presence of these symptoms however other signs of professional burnout have also been widely acknowledged and accepted, including poor patient satisfaction, impaired professionalism and communication, depression and suicidal ideations, professional errors and near misses, sickness and absenteeism [1,2,3,4,5]. Therefore, transcripts from resident–resident representative meetings, PEC meetings, CCC meetings, supervisor reports, and multi-source feedback on clinical care, professionalism and communication from supervisors, peers and patients were analyzed qualitatively for signs of burnout in residents. Resident ACGME-I surveys, and sickness and absenteeism records were also evaluated quantitatively for signs of burnout.

In summary, 20.7% of residents anonymously declared that they were burnt out when asked. The CCC committee found approximately 7% of residents were in difficulty and required remediation. The ACGME-I residents’ survey had a response rate of 72%, but it did not suggest burnout was 62.1% as reported by the aMBI; 92% of respondents did not violate their duty hours (<80 h per week); 77% felt they could raise concerns without fear; 27% felt their residency program was “good”, while 62% felt their residency was “great”; 89% had a positive experience of their residency program. The median number of sick days per annum was 2 (IQR: 1–3.5 days), with sick leave and absenteeism high among a small proportion of residents, 24.1% (4–15 days).

A review of the qualitative data collected from the resident-resident representative interviews and PEC meetings revealed burnout, and terms relating to it, “stress” or “discomfort”, were raised on three occasions. On the first occasion, conflict of daily schedules and disagreements that ensued with senior anesthetists were causing residents to “feel uncomfortable” e.g., departmental teaching occurring simultaneously with service provision duties. Most residents felt uncomfortable asking their seniors to be excused from clinical duties to attend teaching and did not want to be perceived as “self-entitled” (descriptor used by resident-representatives). The second occasion was regarding the inability to obtain leave from work to study for postgraduate examinations. Resident representatives voiced concerns such as: 

“Our mental or emotional health would be better if we are allowed to take leave for exam preparations.”

The third occasion was regarding perceived awkwardness in giving and receiving feedback from faculty. No concerns about a high prevalence of burnout were raised in the review of multiple data sources. These, together with the MBI-HSS results, strongly suggested the prevalence of burnout was low.

### 3.5. The aMBI Overestimated the Prevalence of Burnout

After we established with a high degree of confidence that burnout in our anesthesiology program was low, receiver operating characteristic (ROC) curves (Figure 2) were plotted for MBI-HSS and the aMBI using “clinical diagnosis” as the gold standard (20.7%). The performance statistics of the MBI-HSS and the aMBI were calculated as shown in Table 3. The aMBI was found to have a low AUROC score of 0.74 (95% CI: 0.61–0.85) and a very low positive predictive value of 33.3% (95% CI: 27.5–39.8%), therefore the high false-positive rate (52.2%) of the aMBI can lead to Type 1 errors and overestimation of the prevalence of burnout, which would be amplified in large population studies. Logistic regression of demographic details of anesthesiology residents (independent variables) and the presence of burnout (dependent variable) did not detect any significant relationships.

### 3.6. Stressors in Singapore Anesthesiology Residents

Long working hours (86.4%), inability to pursue personal interests (76.9%), difficult colleagues or seniors (76.9%), fear of making mistakes (76.9%), fear of being reprimanded (76.9%), lack of control over working patterns (76.9%) and nature of the specialty (76.9%) were frequently perceived stressors in burnt out anesthesiology residents. In anesthesiology residents without burnout, insufficient time to study for exams (62.1%), difficult colleagues or seniors (55.2%), postgraduate examinations (53.4%), fear of making mistakes (51.7%), fear of being reprimanded (50.0%), lack of control over working patterns (50.0%), inability to pursue personal interests (50.0%) and nature of the specialty (50.0%) were frequently perceived as stressors. Figure 3 summarizes the frequency of perceived stressors in burnout and non-burnout residents. Logistic regression did not reveal any relationship between perceived stressors and the presence of burn out.

## 4. Discussion

Burnout is a complex multi-dimensional syndrome, and a multitude of diagnostic tools have been developed to diagnose it. These include, but are not limited to, the Maslach Burnout Inventory, Copenhagen Burnout Inventory [29], Oldenburg Burnout Inventory [30], and the Physician Work-Life Study’s Single Item (embedded in “Mini-Z”) [31]. Nonetheless, a universal consensus on which diagnostic tool should be used to measure burnout, and the diagnostic criteria of burnout is currently lacking. Consequently, there is significant heterogeneity observed in systematic reviews and meta-analyses that attempt to accurately report the prevalence of burnout in clinicians [14]. Even if the scope of a systematic review is limited to the field of anesthesiology alone, identical problems with heterogeneity are encountered and for the same reasons [32]. Therefore, the MBI-HSS, being the most robust and validated tool to evaluate burnout in clinicians, was chosen as the gold standard tool for this study [33].

However, we acknowledge that even for the MBI-HSS tool, multiple criteria of burnout have been used. In the first edition of the MBI, the criterion for burnout was defined as the presence of high-risk EE scores (≥27), high-risk DP scores (≥13) and high-risk PA scores (≤33) [34]. This criterion was adopted by most studies that use the MBI-HSS to study burnout out in physicians [14]. However, the cut-off for high-risk PA has been revised to PA ≤ 31 in the third edition of the MBI-HSS. Due to concerns of false negatives, under detection, and easier user administration, variations of the diagnostic tool and diagnostic criteria have also been proposed and adopted by other researchers. High EE risk, high DP risk, but not high PA risk, was reported to be equally predictive of burnout when compared to the MBI-HSS [35,36]. Nonetheless, Maslach supported the definition of burnout as the presence of either “high-risk EE scores and high-risk DP scores” or “high-risk EE scores and high-risk PA scores” [22]. Therefore, this criterion was chosen to define burnout when using the MBI-HSS for the purposes of this study.

Within anesthesiology, the MBI-HSS is the most commonly used tool internationally to study burnout (9/15 studies) [32]. However, recently this trend has shifted toward the aMBI, especially in the US [9,11,17,18,19,20,21]. Our graduate medical education was previously modeled after the UK system, but since 2010, Singapore has collaborated with the US ACGME-I to establish its residency programs for specialist training in anesthesiology. Therefore, it was important to compare directly the prevalence of burnout in our new residency program to the prevalence of burnout in US anesthesiology residents. This was one of the rationales for using the aMBI as a second diagnostic tool. The aMBI was first proposed by Gabbe et al. (2002), defining burnout as the presence of two or more moderate risk scores on any subscale [16]. This was validated in great depth by the authors but lacked rigorous validation by other researchers. Despite this, and given that most large-scale US studies of anesthesiology residents used the aMBI, such a significant statistical difference between the MBI tools was not anticipated. Subsequently, after rigorous analyses, we rejected the null hypothesis that the MBI-HSS, and the aMBI, produced similar estimations of burnout.

The prevalence of burnout in US anesthesiology residents was previously reported at 51% using the aMBI [9]. However, we report with a high degree of confidence that the prevalence of burnout in our anesthesiology program is approximately 20.7% to 22.4% and not 62.1% as reported by the aMBI. Our results are similar to the prevalence of burnout reported in anesthetic trainees in the UK (25%) [37]. However, we were unable to meaningfully test for a difference between burnout proportions in Singapore and US anesthesiology residents because of the limitations of the aMBI tool as shown in this study. The only permissible comparison that could be made is the proportion of residents who had moderate to high-risk scores in two or more subscales using the aMBI, which demonstrated no statistical difference between Singapore and US anesthesiology residents; 62.1% vs. 51.1%, respectively (*p* = 0.09). One study reported the prevalence of burnout in medical and surgical specialties in Singapore using the MBI-HSS, but because a non-validated criterion of burnout was used, the prevalence was estimated at 80.7% [38]. This is of doubtful significance since other limitations in the study were identified [39]. Consequently, meaningful comparisons between our anesthesiology residents with other medical and surgical specialties in Singapore was not possible.

In the present study, there was good correlation between the aMBI and the MBI-HSS subscales scores. However, the modified subscale cut-offs and diagnostic criteria used by the aMBI led to the overestimation of burnout in this study. If the Maslach-recommended criterion of burnout is used with the aMBI tool, the estimated prevalence rate in our cohort would have been 20.7% (12/58). In this example, there would have been no statistical difference in burnout detection between both tools (22.4% vs. 20.7%, *p* = 0.82). Further research is required to investigate whether this is a suitable approach.

We are of the opinion that the correct diagnosis of burnout syndrome is more important than the overestimation of burnout within resident cohorts. Resources for trainee support are often finite and can be sub-optimal in institutions lacking funds and manpower. Using this study as an example, the aMBI tool would have overestimated burnout in our residents at 62.1%. However, it would not have been feasible for us to offer remediation and support services to more than half of our cohort, which would have proved unnecessary and it would have had a negative impact on clinical training and service provision downstream. It could have also provoked skepticism among clinical supervisors and institutional stakeholders. Therefore, it is a better and more sustainable approach to channel support to trainees who have severe burnout and require it the most. Given the limitations of current diagnostic tools, a more appropriate alternative to overestimating burnout would be to use the MBI-HSS with stringent criteria of burnout to screen residents and obtain a conservative estimate. Correlation of test scores with the “clinical picture” from multiple sources employed herein can then be applied effectively to a much smaller group of residents to accurately identify the presence of burnout. We acknowledge the MBI-HSS is a lengthy survey which can impact response rates, therefore, there should be further development of an alternative screening tool or improvement of the 12-question aMBI.

Addressing the causes of burnout is important. Personal management strategies include physical exercise, hobbies, meditation, spiritual strategies, social support, and communication [2]. At an institutional level, limited interventions and safeguards have been implemented globally [1]. These include working-time restrictions (e.g., 80 working hours per week restriction as per ACGME-I requirements), protected rest periods during and between shifts, and counseling services. Mindfulness courses, resilience courses, and the provision of mentoring to trainees early in their career could be impactful in tackling burnout, but research in these areas is ongoing [40,41,42].

Long hours, lack of control over working patterns, postgraduate exams, and medico-legal issues are frequently perceived stressors in the US and UK [9,43,44]. This study observed similar issues in Singapore anesthesiology residents. Unexpectedly, difficult colleagues, fear of making mistakes, and being blamed were frequently perceived stressors in burnt out and non-burnt out residents, which may have been due to a combination of organizational and cultural differences. Nonetheless, improving trainee-trainer relationships and minimizing burnout by creating a “no blame” culture is an important area for development in our organization. 

There are several limitations to this study. Firstly, even though we achieved a response rate of 67.4%, there remains a possibility of non-response bias. Secondly, the total number of respondents was small (*n* = 58), therefore the prevalence reported herein cannot be directly extrapolated to different institutions and countries. However, the small cohort size enabled resident representatives and the research committee to interview most, if not all, the residents in our residency program to validate the results derived by the MBI tools. This is an important limitation encountered by larger studies. More importantly, the statistical analyses remain robust, and a larger cohort size would not improve the performance of the diagnostic tools that were used. Thirdly, even though the qualitative section of our survey on stressors offered free text responses, almost all responses by residents were dichotomous. Therefore, the depth of information extracted from the questions may be limited. Finally, although the MBI-HSS has been extensively validated and used in evaluating burnout among healthcare professionals, its limitation as a diagnostic tool should be recognized because burnout syndrome itself is poorly characterized.

## 5. Conclusions

In summary, this study established that burnout in Singapore anesthesiology residents is 20.7% to 22.4%, which is similar to the prevalence reported in UK anesthesiology trainees. More significantly, it was demonstrated that burnout in anesthesiology residents could be detected accurately using the 22-question MBI-HSS, however false positive rates with the 12-question aMBI were high, which led to the overestimation of burnout. Further research to improve the diagnostic criteria of the 12-question aMBI is required, and until then, caution should be taken in the interpretation of its results.

## Figures and Tables

**Figure 1 jcm-09-00061-f001:**
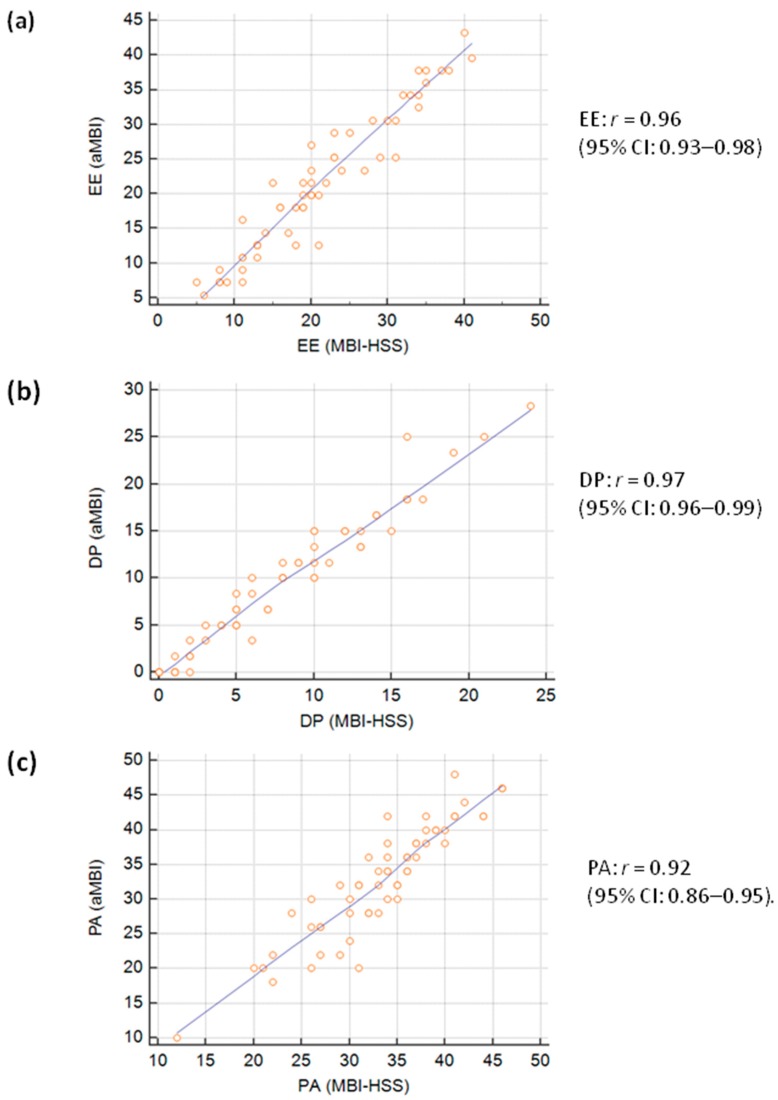
Scatter plots demonstrating the correlation of (**a**) emotional exhaustion (EE), (**b**) depersonalization (DP), and (**c**) personal achievement (PA) scores between the Maslach Burnout Inventory for healthcare professionals (MBI-HSS) and abbreviated MBI (aMBI).

**Figure 2 jcm-09-00061-f002:**
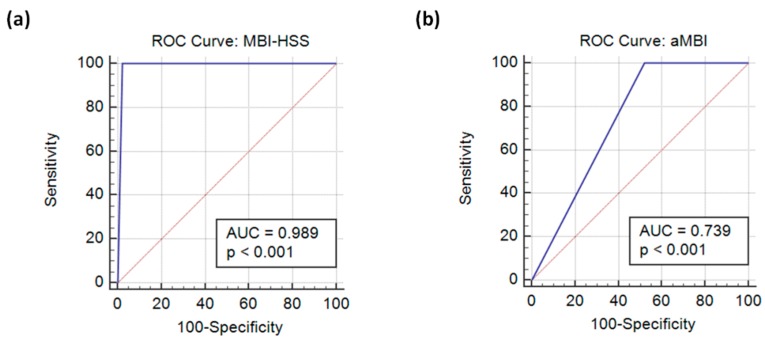
ROC curves for the (**a**) MBI-HSS and the (**b**) aMBI.

**Figure 3 jcm-09-00061-f003:**
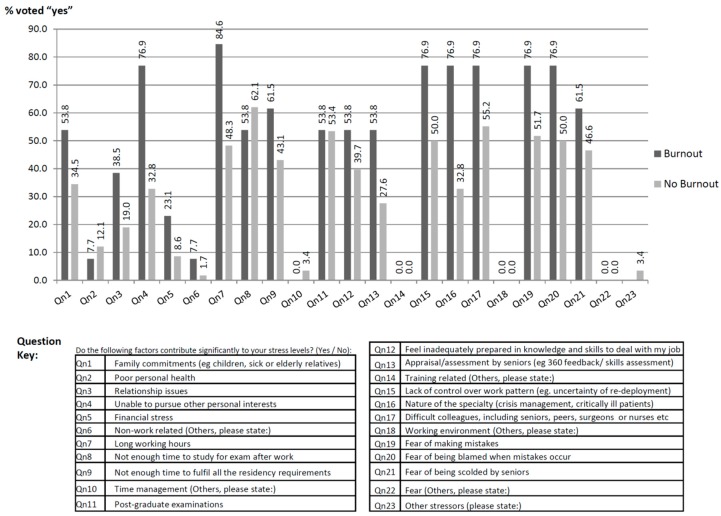
Frequency of stressors reported in burnt out and non-burnout anesthesiology residents.

**Table 1 jcm-09-00061-t001:** Demographics of respondents in relation to the entire cohort of residents in the anesthesiology residency program. * BC denotes board certification or certification for completion of training (CCT).

Year of Training			Total Cohort Size *n* = 86 (a)	Respondents, *n* = 58 (% of Cohort) (b)	Chi Square Test (a) vs. (b)
Training Grade	US Equivalent	UK Equivalent			
JR1	CAT-1	ST3	15	11 (73.3)	*p* = 0.65
JR2	CAT-2	ST4	17	14 (82.4)	
JR3	CAT-3	ST5	21	12 (57.1)	
JR4	Fellowship	ST6	7	4 (57.1)	
SR1	Fellowship/BC *	ST7	17	8 (61.5)	
SR2	Fellowship/BC *	Fellowship/BC *	9	9 (100)	
**Gender**			**Total cohort size**	**Respondents** **(% of cohort)**	**Chi square test** **(a) vs. (b)**
Males			28	17 (60.7)	*p* = 0.68
Females			58	41 (70.7)	
**Age Groups**			**Total cohort size**	**Respondents** **(% of cohort)**	**Chi square test** **(a) vs. (b)**
<25			0	0	*p* = 0.99
25–28			25	17 (68.0)	
29–32			38	27 (71.1)	
33–36			20	12 (60.0)	
>36			3	2 (66.7)	
**Country of graduation (medical school)**			**Total cohort size**	**Respondents** **(% of cohort)**	**Chi square test** **(a) vs. (b)**
National University of Singapore (NUS, YLL)			49	29 (59.1)	*p* = 0.71
Duke-NUS, Singapore			8	6 (75.0)	
Overseas			29	23 (79.3)	

**Table 2 jcm-09-00061-t002:** Comparisons of subscale cut-off values and burnout symptoms in the MBI-HSS and the aMBI. Parametric data reported as mean ± standard deviation (SD); non-parametric data as median and interquartile range (IQR).

Subscales	MBI-HSS Risk Stratification by Scores	MBI-HSS Results by Respondents (%)	MBI-HSS Subscale Scores	aMBI Risk Stratification by Scores	aMBI Results by Respondents (%)	aMBI Subscale Scores
EE	High: ≥27	17 (29.3%)	Median = 20.0 IQR: 13–29 (Mean = 21.0)	High: ≥27	17 (29.3%)	Median = 19.8 IQR: 12.6–28.8 (Mean = 21.3)
	Moderate: 17–26	20 (24.5%)		Moderate: 19–26	15 (25.9%)	
	Low: 0–16	21 (36.2%)		Low: 0–18	26 (44.8%)	
DP	High: ≥13	13 (22.4%)	Median = 7.0 IQR: 2–12 (Mean = 7.7)	High: ≥10	29 **(50%)**	Median = 9.2 IQR: 3.3–15.0 (Mean = 9.1)
	Moderate: 7–12	17 (29.3%)		Moderate: 6–9	6 (10.3%)	
	Low: 0–6	28 (48.2%)		Low: 0–5	23 (39.7%)	
PA	High: 0–31	20 (34.5%)	Mean = 33.5 ± 7.0	High: 0–33	28 **(48.2%)**	Mean = 32.9 ± 8.3
	Moderate: 32–38	25 (43.1%)		Moderate: 34–39	15 (25.9%)	
	Low: ≥39	13 (22.4%)		Low: ≥40	15 (25.9%)	
Note:	MBI-HSS criteria for burnout: High risk EE scores + High risk DP scores or High risk EE scores + High risk PA scores Burnout detected: 22.4% (13/58)			aMBI criteria for burnout: Moderate risk in 2 or more subscales Burnout detected: 62.1% (36/58)		

**Table 3 jcm-09-00061-t003:** Performance statistics of the MBI-HSS and aMBI as diagnostic tools. Results displayed with 95% confidence intervals (CI).

Parameter	MBI-HSS (95% CI)	aMBI (95% CI)
Sensitivity (%)	100.0 (73.5–100)	100.0 (73.5–100.0)
Specificity (%)	97.8 (88.5–99.9)	47.83 (2.9–63.1)
AUROC score	0.99 (0.92–1.00)	0.74 (0.61–0.85)
Positive likelihood ratio	46.0 (6.6–319.6)	1.9 (1.5–2.5)
Negative likelihood ratio	0 (−)	0 (−)
Positive Predictive Value (%)	92.3 (63.3–98.8)	33.3 (27.5–39.8)
Negative Predictive Value (%)	100.0 (−)	100.0 (−)
Accuracy (%)	98.3 (90.8–100.0)	58.6 (44.9–71.4)

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
