# Peer review of "The Abbreviated Maslach Burnout Inventory Can Overestimate Burnout: A Study of Anesthesiology Residents"

_jcm, 2019, doi:10.3390/jcm9010061_

Round 1
Reviewer 1 Report
I read the manuscript.
Generally, this study is well organized and written.
However, I also think that there are several issues to be approached by the authors.
# Title
It seems to me that the title is misleading. I recommend the authors should reconsider the title to emphasize the finding of this study positively.
# Abstract
The abstract contains details of the statics. I think those can be omitted.
# English language
I think the English language can be improved.
Reviewer 2 Report
It is a very interesting topic.
A series of elements are presented below in case they can be useful to improve some aspects of the manuscript:
- With regard to the title, it should be taken into account that, although it is true that it gives a lot of information, it has a very long extension. In addition, it ends with an endpoint. An endpoint is not required in a title.
- With respect to the summary, it would be convenient to introduce the number of women or men who participate in relation to the total number of participants. Besides, the number of participants is low. The sample value should be increased. Use synonyms to provide more fluency to certain sentences where the same word is repeated. For example, determine.
- Concerning Introduction, in line 130 the sentence which starts with "4" should be changed thus it is convenient to use a word instead of a number at the beginning of a sentence. At the end of the introduction, general objectives and specific objectives should be described. It would also be interesting to include hypotheses to later accept or reject these hypotheses.
- The methodology is widely described.
- The discussion is also widely described. In this part, it should be interesting to explicit if the starting hypotheses are confirmed. For this, it would be necessary to expose these hypotheses previously. Regarding the quotes, there is an important number of new quotes that could be used in the introduction of the manuscript. So, there will be a stronger connection between the introduction and the discussion.
- Eventually, regarding the references, the authors have been used in the manuscript recent articles.
To sum up, the manuscript has got a good structure. Maybe, the theoretical framework could be improved. Another drawback is the sample size. When studying such a specific group it is difficult to obtain a lot of participants but at the same time, when the sample is reduced, it loses external validity.
Round 2
Reviewer 2 Report
Thank you very much for your great work and effort.